# RepoFusion: Training Code Models to Understand Your Repository

## Abstract

Despite the huge success of Large Language Models (LLMs) in coding assistants like GitHub Copilot, these models struggle to understand the context present in the repository (e.g., imports, parent classes, files with similar names, etc.), thereby producing inaccurate code completions. This effect is more pronounced when using these assistants for repositories that the model has not seen during training, such as proprietary software or work-in-progress code projects. Recent work (Shrivastava et al., 2022; Zhang et al., 2023) has shown the promise of using context from the repository during inference. In this work, we extend this idea and propose *RepoFusion*, a framework to train models to incorporate relevant repository context. Experiments on single-line code completion show that our models trained with repository context significantly outperform much larger code models as CodeGen-16B-multi ($\sim 73\times$ larger) and closely match the performance of the $\sim 70\times$ larger StarCoderBase model that was trained with the Fill-in-the-Middle objective. We find these results to be a novel and compelling demonstration of the gains that training with repository context can bring. We carry out extensive ablation studies to investigate the impact of design choices such as context type, number of contexts, context length, and initialization within our framework. Lastly, we release a dataset with three types of repository contexts to facilitate further research in this domain.

## 1 Introduction

Large Language Models (LLMs) of code (Svyatkovskiy et al., 2020; Chen et al., 2021; Fried et al., 2022; Wang et al., 2021; Nijkamp et al., 2023; Li et al., 2022; Allal et al., 2023) have gained significant popularity. The demand for these models has further increased with their integration into code assistants like GitHub Copilot[1] and TabNine[2], and their popularity is anticipated to grow further as more developer-assistance products are developed around them.

Despite their remarkable capabilities, LLMs of code often struggle to generalize effectively in unforeseen or unpredictable situations, resulting in undesirable predictions. Instances of such scenarios include code that uses private APIs or proprietary software, work-in-progress code, and any other context that the model has not seen while training. To address these limitations, one possible approach is to enhance the predictions of these models by incorporating the wider context available in the repository. Leveraging the structure and context of the repository can take into consideration dependencies between files, such as imports and parent classes, and provide valuable insights into coding patterns that may be specific to the organization or user. Recent works (Shrivastava et al., 2022; Zhang et al., 2023; Ding et al., 2022) have shown promising results in utilizing repository-level context in conjugation with LLMs of code. It was also shown in Shrivastava et al. (2022) that without specialized training, it is challenging to integrate multiple relevant contexts from the repository. Building upon these findings we propose RepoFusion, a training framework for learning to combine multiple relevant contexts from the repository in order to generate more accurate and context-aware code completions.

In this work, we focus on the task of single-line code completion (Hellendoorn & Devanbu, 2017; Shrivastava et al., 2020) which simulates real-world scenarios where users are editing existing files in

---

[1]https://github.com/features/copilot/
[2]https://www.tabnine.com/

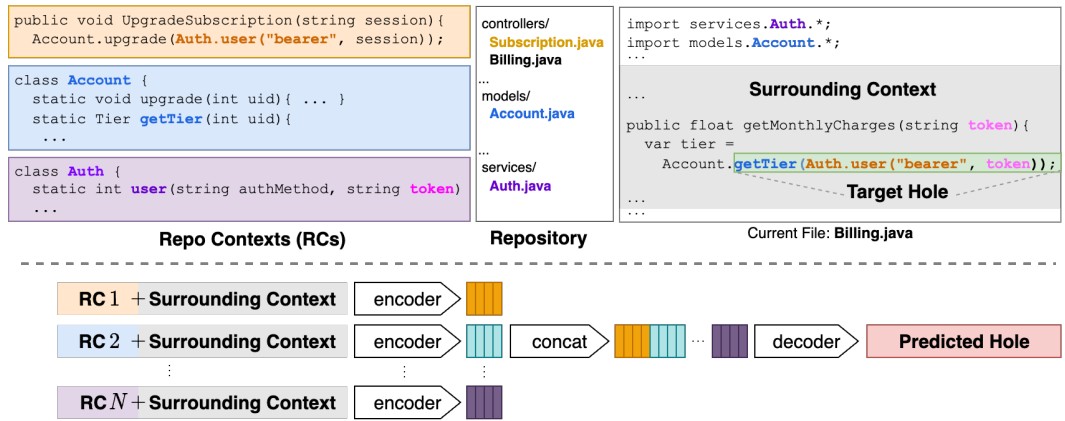

Figure 1: Given multiple relevant contexts from the repository (Repo Contexts), RepoFusion appends the Surrounding Context (highlighted in gray) to each repo context, encodes them separately, and combines them to produce a prediction of the target hole.

an IDE. With reference to Figure 1, this means that we have to predict the missing section, referred to as the *target hole* (highlighted in green), starting from the cursor's position until the end of the line. We see that the completion of this target hole will benefit from context not just in the current file (the variable name `token`), but also from other files in the repository. Specifically, the context from the imported file Account.java provides insight into the usage of the `getTier` method, while the sibling file Subscription.java offers guidance on the usage of `Auth.user("bearer"`, with the definition of `Auth` found in the imported file Auth.java. Given these relevant code snippets from across the repository which we call *Repo Contexts (RCs)*, RepoFusion uses the Fusion-in-Decoder (Izacard & Grave, 2021) architecture to combine these. Specifically, each repo context is appended with the *surrounding context* i.e., a window around the target hole excluding the target hole (highlighted in gray) and encoded separately. A decoder jointly attends to the concatenated encoded representations to produce a prediction for the target hole (highlighted in red). The key contributions of our paper can be listed as follows:

- We propose RepoFusion, a framework that helps code models to make better predictions by learning to combine relevant contextual cues from the repository.
- Through extensive experiments we establish that RepoFusion, a 220M parameter model, significantly outperforms several larger models trained on the next-token prediction objective such as CodeGen-16B (Nijkamp et al., 2023). Furthermore, despite being approximately 70 times smaller in size, our model closely matches the performance of StarCoderBase (Li et al., 2023), a 15.5B parameter LLM trained with the Fill-in-the-Middle (Bavarian et al., 2022) objective.
- We conduct thorough ablation studies to gain insights into the key factors influencing RepoFusion, such as the nature of repository contexts, their lengths, the number of repository contexts, and other training configurations. One of the crucial findings is that leveraging information from diverse sources within a repository is the key to RepoFusion's effectiveness.
- We create and release *Stack-Repo* [3], a dataset of 200 Java repositories with permissive licenses and near-deduplicated files that are augmented with three types of repository contexts.

## 2 TRAINING WITH REPOSITORY CONTEXT

In this section, we briefly describe Fusion-in-Decoder (Izacard & Grave, 2021), the repository contexts we used, and the details of our RepoFusion framework.

---

[3]https://huggingface.co/datasets/anonymousTheStackRepo/the-stack-repo

## 2.1 FUSION-IN DECODER

Fusion-in-Decoder (Izacard & Grave, 2021) (FiD) is a method to train a language model to combine information coming from multiple sources. In the original work, FiD was used for open-domain question answering. In the FiD approach to question answering, a sequence-to-sequence model takes support passages concatenated with the question as inputs and produces an answer as the output. Each support passage is appended to the question and encoded independently by the encoder. The encoded representations are then concatenated and fed to the decoder which jointly attends to them to produce the answer. In this work, we adapt FiD for the setting of code completion.

## 2.2 REPOSITORY CONTEXTS

Motivated by the syntax and semantics of programming languages as well as the common coding patterns, Shrivastava et al. (2022) proposed a set of repo-level prompt proposals that leverage the structure and the relevant context in files across the repository. A prompt proposal (PP) is a function that takes in the target hole's location and the associated repository as input and returns a string called *Prompt Proposal Context (PPC)* as output. The prompt proposal context is created by extracting a particular type of context (prompt context type) from a particular category of related source files (prompt source). Examples of prompt sources are the current file, files that are imported into the current file, files that have a similar name as the current file, etc. Examples of prompt context types are lines following the target hole, method names and bodies, identifiers, string literals, etc. Combining these prompt sources and prompt context types gives us a total of 63 prompt proposals (see Appendix B.4 of Shrivastava et al. (2022) for details). It should be noted that the context from the beginning of the current file up to the position of the hole, as well as the context following the target hole within the current file are also types of prompt proposal contexts. We will refer to these as the prior PPC (or just *prior*) and the post PPC (or just *post*), respectively in the remainder of the paper. Note that depending on the target hole, some prompt proposal contexts might be empty (e.g. if the target hole is towards the very beginning of the file, there might not be any import statements from the current file to get context from).

Repo-level prompt proposals can be thought of as a deterministic retrieval mechanism that returns the relevant code snippets from the repository. To understand the role of the retrieved repo contexts, apart from prompt proposals, we also consider two other mechanisms for retrieving repository-level context (see Appendix for implementation details): (a) BM25: The context from each file in the repository is scored using BM25-based (Jones et al., 2000) similarity with the surrounding context, and (b) RandomNN (also used in Shrivastava et al. (2022)): From a list of randomly selected chunks from the repository, we select the top-k based on the similarity of the embedded chunks with the embedded surrounding context in the representation space.

## 2.3 REPOFUSION

The core idea of RepoFusion is to train a code model to be aware of the context in the repository such that it helps in generating an accurate prediction of the target hole. Given a set of retrieved repo contexts, RepoFusion learns to combine the relevant parts of these contexts using the FiD approach as described in Section 2.1. The surrounding context is concatenated with each repo context and then encoded independently (see Figure 1, bottom). Note that for our purpose, since we want the code model to complete the target hole, we append the surrounding context toward the end of the repo context. This is different from the original work (Izacard & Grave, 2021), where the question (analogous to the surrounding context in our setting) is appended at the beginning of each passage (analogous to the repo context in our setting). RepoFusion uses $N$ repo contexts of length $l$ tokens each. We experimented with the following four strategies for producing and ordering the repo contexts based on the prompt proposal contexts (see Figure 2).

1. **Truncated-Ranked (T-Rank):** In this setting, one prompt proposal context yields one repo context. We truncate each prompt proposal context (i.e., take only the first $l$ tokens) to form the respective repo context and discard the rest. The repo contexts are ordered based on the ranking of the prompt proposals [4] on the validation split of the Google Code archives

---

[4] https://github.com/shrivastavadisha/repo_level_prompt_generation/blob/main/get_info_from_hole_predictions.py

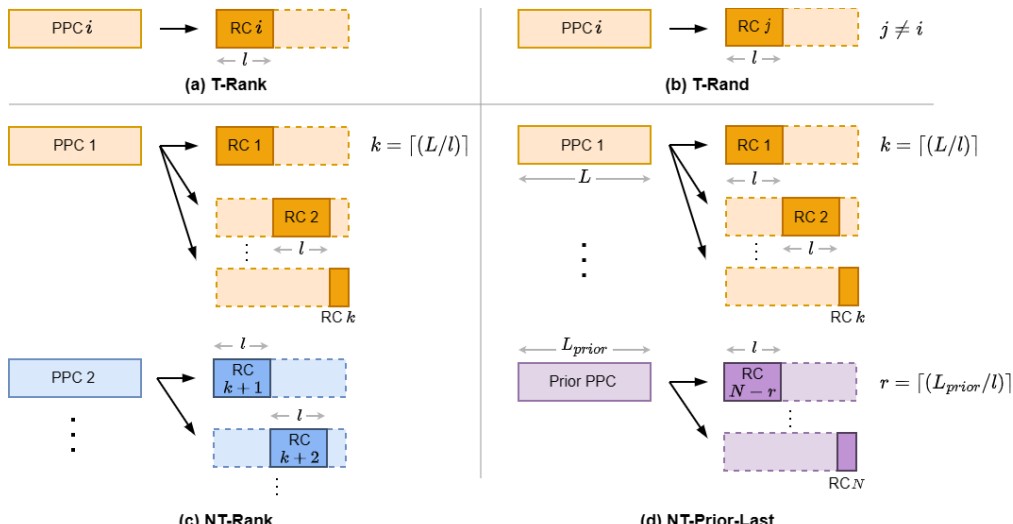

Figure 2: Different strategies employed for producing repo contexts (RCs) from prompt proposal contexts (PPCs): **(a) T-rank:** we truncate the $i$-th ranked PPC to yield the $i$-th RC. **(b) T-rand:** we position the truncated $i$-th PPC at a random position $j$ in RepoFusion's sequences of RCs. **(c) NT-Rank:** each PPC yields as many RCs as necessary to exhaust all of its tokens without truncation. **(d) NT-Prior-Last:** we reserve the last $r$ RCs for the Prior PPC and fill the rest RCs as in NT-Rank.

> dataset of (Shrivastava et al., 2022). Given that our work and Shrivastava et al. (2022) both target Java, it seemed reasonable to us to directly use this ordering.
> 2. **Truncated-Random (T-Rand):** Same as T-rank except that the repo contexts are ordered randomly. This helps us understand the role of the specific ranking of PPs from (Shrivastava et al., 2022).
> 3. **Not Truncated-Ranked (NT-Rank):** The prompt proposals are ranked based on the same order as in T-Rank. Unlike T-rank, here we avoid the truncation of prompt proposal contexts. Instead, we construct as many repo contexts from each prompt proposal context as necessary, namely a PPC of length $L$ will contribute $k = \lceil (L/l) \rceil$ RCs. We then proceed to the next in order prompt proposal and continue so until we have selected $N$ repo contexts. Unlike T-rank, this setting allows RepoFusion to see the entirety of top-ranked prompt proposals at the cost of potentially ignoring the lower-ranked ones.
> 4. **Not Truncated-Prior-Last (NT-Prior-Last):** Same as NT-Rank except that the prior PPC is always ordered at the end. Since the decoder attends to the concatenated encoded representations of the repo contexts in the same order as it is presented as inputs, this strategy helps us understand the role of continuing code generation from the encoded representation of the prior PPC as the most recently attended representation in the decoder. Note that depending on the value of $N$, it may be necessary to remove certain chunks of top-ranked PPCs in order to accommodate the prior PPC at the end.

Similar to Izacard & Grave (2021), we format each repo context with special tokens to mark the beginning of the surrounding context and the repo context, as well as for the name of the repo context (which is the same as the name of the prompt proposal). Please see the Appendix for details on these tokens and other architectural details of RepoFusion.

## 3 EXPERIMENTS AND RESULTS

In this section, we describe the process of creation of our dataset Stack-Repo and the details of experiments. We then present the results of evaluating RepoFusion and other models on the test set. This is followed by presenting the findings of extensive ablation studies carried out on the validation set to gain deeper insights into the individual contributions of each component in our framework.

## 3.1 DATASET CREATION

In this work, we build upon a modified version of The Stack V1.1 (Kocetkov et al., 2022). The modified version [5] consists of near-deduplicated code repositories with permissive licenses from GitHub. For our experiments, we take only the Java subset (files with .java extension) of this dataset.

**Creation of Target Holes:** For creating target holes needed for training and evaluating RepoFusion, we choose a set of repositories randomly from the Java subset of the Stack and divide them into training, validation, and test splits in the ratios 2:1:1 respectively. We only consider repositories that contain at least 20 near-deduplicated files. For each repository, we choose target holes from every code line (excluding comments in natural language and blanks) in all the files. In order to tokenize a code line, we used common Java code delimiter tokens [6]. We chose a random token within the line and the rest of the line starting from that position till the end constitutes the target hole. By not selecting target holes based on the tokenizer of a specific code model, we can ensure that the tokenizer remains unbiased and does not implicitly favor any particular code model in our experiments. To avoid bias from large repositories while training, we cap the maximum contribution of target holes from a repository to 10000, i.e. if the total number of holes in the repository exceeds 10000, we select 10000 holes randomly from the total holes. Please see Table 1 for the statistics of Stack-Repo.

Table 1: Statistics of Stack-Repo

| Feature | Train | Val | Test |
|---|---|---|---|
| **# Repositories** | 100 | 50 | 50 |
| **# Files** | 20310 | 11172 | 13202 |
| **# Holes** | 435890 | 220615 | 159822 |

**Creation of Repo Contexts:** For each target hole, we use the implementation [7] from (Shrivastava et al., 2022) to extract prompt proposal contexts. We take two lines above and two lines below the target hole excluding the target hole as the surrounding context. For obtaining the embeddings for RandomNN repo contexts, we use pre-trained CodeBERT (Feng et al., 2020). For constructing the BM25 repo contexts, we use the implementation from the Rank-BM25 package [8]. To improve efficiency, we store the repo contexts for each target hole in advance. Note that even though our target hole and repo context creation strategies have been inspired from Shrivastava et al. (2022), our dataset, Stack-Repo is significantly bigger in size. Apart from code completion, Stack-Repo can serve as a benchmark for various other code-related tasks involving repository context, such as bug repair and pull request resolution. We plan to release it under the same license as The Stack (Kocetkov et al., 2022) to support future research in these areas.

## 3.2 EXPERIMENTAL DETAILS

**Training of RepoFusion:** We use the 220M parameter CodeT5-base (Wang et al., 2021) encoder-decoder model as our base code model for RepoFusion. We found that the pre-trained CodeT5 model was not good at completing Java code (see Appendix for initial results). Therefore, to obtain a base model for RepoFusion training we finetuned CodeT5-base with an input context length of 512 using the next-token prediction objective on Java repositories from the dataset described in Section 3.1. Specifically, we used the repositories that were not included in Stack-Repo. For each file, we randomly sample ten pivot points with the code context prior to the pivot location in the file serving as the input to the encoder of CodeT5. The finetuned CodeT5-base model was then used to initialize the training of RepoFusion. Based on the validation set performance, we found that for RepoFusion, NT-Prior-Last with $N = 32$ and $l = 768$ works the best. We provide complete details of training RepoFusion and finetuning CodeT5 in Appendix.

**Baselines:** To benchmark the performance of RepoFusion, we conducted experiments with several methods, with each model utilizing the recommended tokenizers specific to the method and employing

---

[5]https://huggingface.co/datasets/bigcode/the-stack-dedup

[6]`[., (,), [,] , ,{,},,,:,",;]`

[7]https://github.com/shrivastavadisha/repo_level_prompt_generation

[8]https://pypi.org/project/rank-bm25/

a maximum token generation limit of 128 per completion. To ensure a thorough analysis, we have incorporated encoder-decoder models as well as decoder-only models of different sizes, with varying context lengths and two different input context types. We present the details of the methods below:

1. **CodeT5 (FT)**: In addition to the previously described fine-tuned (FT) version of **CodeT5-base**, we also finetuned **CodeT5-large** (770M) with a context length of 512. Next, we assessed the performance of these models using input context lengths of 2048 and 4096. The input context was constructed by either considering the prior PPC (*prior*) alone or by concatenating equal lengths of the post PPC (*post*) and prior.
2. **BigCode models:** We experimented with two models released by BigCode [9]. The first model is **SantaCoder** (Allal et al., 2023), which is a 1.1B parameter model which supports a maximum context length of 2048 tokens and the second is the recently released **StarCoder-Base** (Li et al., 2023) model which is a 15.5B parameter model that can support up to 8192 tokens. Both of these models are trained with Fill-in-the-Middle (Bavarian et al., 2022) (FiM) objective on versions 1.1 and 1.2 of The Stack (Kocetkov et al., 2022), respectively. These models were evaluated with both the prior and post+prior contexts as inputs. For experiments with post+prior, we used the FiM special tokens that were used while training these models. Since these models have been trained specifically to see the post PPC as suffix, they help us understand the role of training with multiple repo contexts in the way proposed by RepoFusion.
3. **CodeGen (Nijkamp et al., 2023):** CodeGen is a decoder-only transformer-based autoregressive model trained with the next-token prediction objective. It supports a maximum context length of 2048 tokens. We experimented with three pre-trained variants of CodeGen, namely **CodeGen-2B-multi**, **CodeGen-6B-multi**, and **CodeGen-16B-multi**. As before, we tried the scenarios where the input context consists of the post + prior as well as when the input context consists of just the prior. These models help us understand the performance of large pre-trained models that are not trained with repo context.

It is important to note that when compared to our RepoFusion model, with the exception of CodeT5-base (FT), all other models are many times larger in size and have been trained on a significantly larger number of tokens. The rationale behind selecting these baselines is to compare the performance of training smaller models with additional repository context against training much larger models without incorporating repository context.

**Evaluation Metric:** We conduct an exact string match between the predicted hole and the target hole, where the predicted hole is the string up to the occurrence of the first newline in the completion. If an exact match is found, it is considered a success; otherwise, it is deemed a failure. We measure the fraction of exact matches over the dataset and call it *Success Rate*.

## 3.3 RESULTS

Table 2 presents the hole completion success rate (along with standard error) in percentage for different methods on our test set, where the standard error is an estimate of the variability in the sample mean of the distribution of exact match. The top two sections of the table display the evaluation results of the finetuned encoder-decoder {CodeT5-base(FT), CodeT5-large(FT)} models and decoder-only {SantaCoder, CodeGen-2B, CodeGen-6B, CodeGen-16B} models, respectively when provided with prior context as input. The table's next two sections present the results of evaluating these models when given with post+prior context as input. In the final section of the table, we showcase the evaluation results of RepoFusion using different effective input context lengths obtained by varying the values of $N$ and $l$.

**Baseline Performance Improves with Model Size and the Addition of Context:** The performance of CodeT5 (FT) models improves as the model becomes bigger (CodeT5-large vs CodeT5-base) and as the input context length increases (2048 vs 4096). We observe a comparable pattern with decoder-only models, where there is a general enhancement in performance as the models grow larger (with a slight indication of saturation) while maintaining a fixed context length. Additionally, we note a substantial improvement in both categories of models when provided with post + prior context as

---

[9]https://www.bigcode-project.org/

Table 2: Completion success rate on the test set for different methods.

| Model | Size (#params) | Effective context length | Context type | Success Rate (%) |
|---|---|---|---|---|
| CodeT5-base (FT) | 0.22B | 2048 | prior | $41.82 \pm 0.12$ |
| CodeT5-base (FT) | 0.22B | 4096 | prior | $46.45 \pm 0.12$ |
| CodeT5-large (FT) | 0.77B | 2048 | prior | $44.73 \pm 0.12$ |
| CodeT5-large (FT) | 0.77B | 4096 | prior | $48.92 \pm 0.12$ |
| SantaCoder | 1.1B | 2048 | prior | $39.51 \pm 0.12$ |
| CodeGen | 2B | 2048 | prior | $49.45 \pm 0.12$ |
| CodeGen | 6B | 2048 | prior | $49.19 \pm 0.12$ |
| CodeGen | 16B | 2048 | prior | $50.20 \pm 0.12$ |
| CodeT5-base (FT) | 0.22B | 2048 | post+prior | $48.89 \pm 0.12$ |
| CodeT5-base (FT) | 0.22B | 4096 | post+prior | $49.97 \pm 0.12$ |
| CodeT5-large (FT) | 0.77B | 2048 | post+prior | $51.72 \pm 0.12$ |
| CodeT5-large (FT) | 0.77B | 4096 | post+prior | $52.43 \pm 0.12$ |
| SantaCoder | 1.1B | 2048 | post+prior | $56.78 \pm 0.12$ |
| CodeGen | 2B | 2048 | post+prior | $53.18 \pm 0.12$ |
| CodeGen | 6B | 2048 | post+prior | $54.03 \pm 0.12$ |
| CodeGen | 16B | 2048 | post+prior | $54.09 \pm 0.12$ |
| RepoFusion ($N = 4, l = 512$) | 0.22B | 2048 | NT-Prior-Last | $65.96 \pm 0.12$ |
| RepoFusion ($N = 8, l = 512$) | 0.22B | 4096 | NT-Prior-Last | $70.38 \pm 0.11$ |
| RepoFusion ($N = 32, l = 768$) | 0.22B | 24576 | NT-Prior-Last | $77.32 \pm 0.10$ |

Table 3: Comparison with StarCoderBase on a test set subset.

| Model | Size (#params) | Effective context length | Context type | Success Rate (%) |
|---|---|---|---|---|
| StarCoderBase | 15.5B | 8192 | prior | $52.97 \pm 0.45$ |
| StarCoderBase | 15.5B | 8192 | post+prior | $79.79 \pm 0.36$ |
| RepoFusion ($N = 16, l = 512$) | 0.22B | 8192 | NT-Prior-Last | $73.67 \pm 0.43$ |
| RepoFusion ($N = 32, l = 2500$) | 0.22B | 80000 | NT-Prior-Last | $78.33 \pm 0.37$ |

input, compared to their respective performances with only the prior context. The SantaCoder model, specifically trained for the FiM task, exhibits the most significant improvement.

**RepoFusion is Effective:** RepoFusion not only exhibits a substantial improvement over its base model (CodeT5-base (FT)) but also surpasses other bigger models, even when utilizing the same effective context length. Furthermore, RepoFusion achieves superior performance compared to the significantly larger CodeGen-16B model, even when constrained to utilize fewer repository contexts to match the effective context length of CodeGen-16B. Furthermore, when provided with additional repo contexts, RepoFusion demonstrates further enhancements in performance.

We also compare RepoFusion with the recently released StarCoderBase (Li et al., 2023) model. StarCoderBase is a 15.5B parameter model which is trained with about one trillion tokens using a FiM objective employing a large input context length of 8192 tokens. The results of this comparison using a random subset of 12500 holes from our test set are depicted in Table 3. Learning to read additional repository context allows RepoFusion to achieve success rate just 1.3% below the performance of the 70 times bigger state-of-the-art StarCoderBase model.

**Prompt Proposals Matter:** The right side of Figure 3 illustrates the success rate of RepoFusion using Random-NN, BM25, and PPC (refer to Section 2.2 for details) when employing T-Rank and NT-Rank. Note that when evaluating Random-NN and BM25, we employed corresponding RepoFusion models

specifically trained to accept Random-NN and BM25 contexts as inputs. The results show that using the repo context from PP (Shrivastava et al., 2022) performs the best.

**The NT-Prior-Last Strategy is Most Effective:** Next, we compare performances of the four different repo context production and ordering strategies that we introduced in Section 2.3. The left side of Figure 3 illustrates the success rate for the four strategies in two distinct settings: $N = 32, l = 768$ and $N = 63, l = 512$. We see that the ordered repo contexts, specifically NT-Prior-Last, NT-Rank, and T-Rank perform better than random ordering of repo contexts (T-Rand). Also, the improved performance of NT-versions when compared to the T-versions, highlights the value of presenting complete context from top prompt proposals, as opposed to displaying truncated contexts from more prompt proposals.

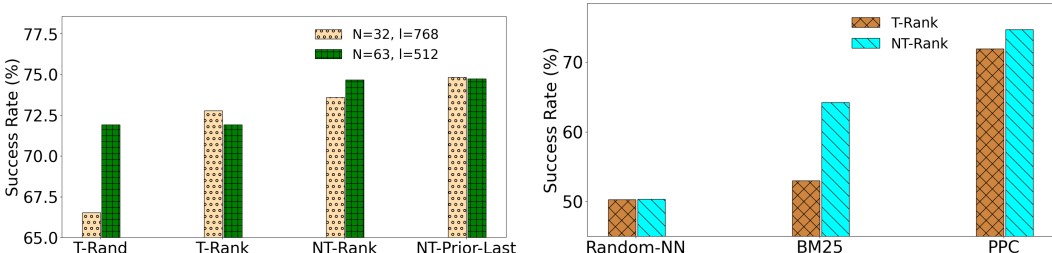

Figure 3: Completion success rate with different approaches to producing repository contexts (RCs). *(Left)* Impact of RC production and ordering strategies; *(Right)* Impact of different RC retrieval methods.

**Longer Repo Contexts are Better:** In the left side of Figure 4, we plot the variation of success rate with different values of the repo context length $l$. For this experiment, we used our best-performing model that was trained using NT-Prior-Last. The results indicate an improvement in the performance with the size of each repo context. However, in both cases ($N = 32, N = 63$), the performance reaches a saturation point as the value of $l$ increases.

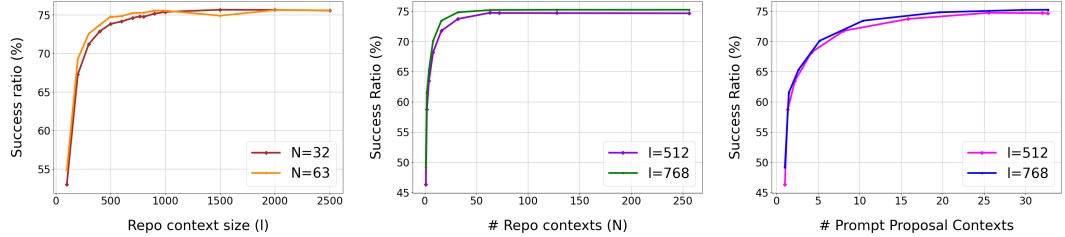

Figure 4: Completion success rate as a function of *(Left)* the length of the individual repo context $l$; *(Middle)* the number of repo contexts $N$; *(Right)* the number of prompt proposal contexts that were used to produce the $N$ repo contexts.

**Using More Repo Contexts is Better:** To understand the role of multiple repo contexts, we evaluated our best-performing model with different values of $N$. We see from the middle part of Figure 4 that the performance of RepoFusion increases upto $N = 63$ when $l = 512$ and up to $N = 32$ with a longer length of repo context $l = 768$. After this, increasing the value of $N$ doesn't lead to further improvements.

**RepoFusion Benefits from Diverse Prompt Proposals:** We additionally look at the success rate as a function of the average number of different prompt proposal contexts that produced the considered repo contexts for each number of repo contexts $N$. Note that the number of PPCs is less than $N$ because often one PPC yields multiple RCs. One can see from the right part of Figure 4 that using many diverse PPCs was essential for getting better performance with RepoFusion.

**Finetuning the Base Model for Next Token Prediction is Important:** Table 4 shows the results of evaluating RepoFusion when the corresponding model is trained by initializing with a pretrained

CodeT5-base model versus initializing with a finetuned version. We observe that while training with repo contexts enhances the performance of the base pretrained CodeT5 model (see Appendix for the performance with pretrained CodeT5), we see that in all cases, there are clear benefits of initializing the training with a model that is finetuned for code completion.

Table 4: Completion success rate when initialized from a pretrained vs finetuned model.

|                | Pretrained       | Finetuned        |
| -------------- | ---------------- | ---------------- |
| **T-Rand**     | 54.67±0.50       | 66.53±0.47       |
| **T-Rank**     | 59.57±0.49       | 72.78±0.45       |
| **NT-Rank**    | 60.88±0.49       | 73.60±0.44       |
| **NT-Prior-Last** | 61.91±0.49    | 74.82±0.43       |

## 4 RELATED WORK

**Information from Outside the Current File:** In the context of source code, harnessing information beyond the current file has been found to be useful. Hellendoorn & Devanbu (2017) utilizes a nested n-gram model with a locality-based cache encompassing all directories in the repository. To capture the structure of the repository, Pashakhanloo et al. (2022b;a) convert it into a relational database and propose a graph-walk mechanism whereas, Lyu et al. (2021) incorporates the API-dependency graph in its LSTM-based code model. While training the kNN-LM (Khandelwal et al., 2020), Xu et al. (2022) incorporates three types of binary variables corresponding to the presence or absence of similar local and global hierarchy. Zhang et al. (2021) leverages the parent class to generate comments for the child class.

**Repository-level Context for Inference in LLMs:** Shrivastava et al. (2022) proposes RLPG, a classifier that selects a prompt proposal based on the target hole and utilizes the context from the chosen prompt proposal and prior context to prompt Codex (Chen et al., 2021). Similarly, RepoCoder (Zhang et al., 2023) iteratively refines the prediction of the target hole by injecting the previous predictions from the LLM in addition to the retrieved context and prior context in the input prompt. In this work, we build upon the insights gained by Shrivastava et al. (2022) regarding the utilization of repository-level information during inference. We extend their findings to various configurations involving different code language models (LLMs), considering a range of context lengths and sizes. Additionally, our framework is trained with context from the repository and learns to effectively leverage multiple relevant contexts sourced from the repository.

**Retrieval-augmented Code Models:** In recent studies (Zhou et al., 2023; Parvez et al., 2021; Lu et al., 2022; Zan et al., 2022; Ding et al., 2022; Borgeaud et al., 2022), attempts have been made to enhance code LLMs by augmenting them with a sparse or dense retrieval mechanism that returns API documentation or relevant code snippets from the repository. The prompt proposals (Shrivastava et al., 2022) used in our work along with BM25 and Random-NN share similarities with these retrieval mechanisms. Note that RepoFusion is independent from the specific retrieval mechanisms employed and thus can seamlessly learn to integrate multiple retrieved contexts, even from different retrieval mechanisms.

## 5 CONCLUSIONS

We propose RepoFusion, a framework that allows training code models with multiple relevant contexts from the repository. By employing RepoFusion in experiments focused on single-line code autocompletion, we highlight the notable enhancements in performance attained through training smaller models with repository context, surpassing the results of training larger models without such context. Deploying RepoFusion, similar to any other code LLMs, requires careful consideration (Chen et al., 2021). The generated code can often be challenging to understand or debug, resulting in developers spending significant time editing and revising the code (Vaithilingam et al., 2022; Mozannar et al., 2022; Barke et al., 2023; Bird et al., 2022). RepoFusion, in combination with the Stack-Repo dataset, opens up exciting avenues for future research in the field of smaller retrieval-augmented LLMs for code. We believe our method can also extend to other code-related tasks such as bug repair, the merging of pull requests, and software documentation/tutorial writing.

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
