# Supplementary Material for "RepoFusion: Training Code Models to Understand Your Repository"

## A    Details on Stack-Repo

For the review process, we have made our dataset available at the link: `https://huggingface.co/datasets/anonymousTheStackRepo/the-stack-repo`. Upon completion of the review process, a de-anonymized version of the dataset will be released under a license similar to that of The Stack (Kocetkov et al., 2022). The details of the license can be found at the Licensing Information section of this page `https://huggingface.co/datasets/bigcode/the-stack`. Stack-Repo consists of 200 near-deduplicated Java repositories (see Table 1 of main paper for details). For each repository within a split (train, validation and test), we provide all files arranged in the directory structure within the repository along with three `.json` files that contain the PP, BM25 and RandomNN repo-contexts. One row of the `.json` file corrsponds to a target hole consisting of the location of the target hole, the target hole as a string, the surrounding context as a string and a list of repo-contexts as strings.

## B    Implementation Details

### B.1    Finetuning CodeT5

As described in Section 3.2 of the main paper, to serve as a better initialization of RepoFusion (also served as a baseline) we finetuned a CodeT5-base model (220M parameters) with an input context length of 512 tokens using the CodeT5 tokenizer. We used an Adam optimizer with Decoupled Weight Decay Regularization (Loshchilov & Hutter, 2019) with weight decay of 0.05 and a learning rate of 4e-05. In addition, we used a linear scheduler with 100 warm-up steps, dropout rate of 0.1, and gradient clipping with a max gradient norm of 1.0. To serve as a baseline, we also finetuned a CodeT5-large model (770 M parameters) with an input context length of 512. We used the same set of hyperparameters for this as mentioned before except that we used a learning rate of 1e-4. The training was carried out on 2 NVIDIA A100 GPUs with memory of 80GB each and a batch size of 32 per GPU for the CodeT5-base model. For CodeT5-large we used 4 A100 GPUs with memory of 80GB each and a batch size of 12 per GPU. The evaluation run was carried out on a single 32GB V100 GPU with a batch size of 32 for CodeT5-base and 48 for CodeT5-large.

### B.2    Training RepoFusion

We use the 220M parameter CodeT5-base (Wang et al., 2021) encoder-decoder model as our base code model for RepoFusion. Our RepoFusion implementation was heavily built on top of the code released by Shrivastava et al. (2022) `https://github.com/shrivastavadisha/repo_level_prompt_generation` (MIT License), as well as the code released by Izacard & Grave (2021) `https://github.com/facebookresearch/FiD` (MIT License). The former was used to obtain repo contexts and the latter was used for the FiD architecture.

Our best RepoFusion model was obtained by initializing the training from a finetuned CodeT5-base checkpoint (see Section B.1 for details). The repo contexts used the NT-Prior-Last strategy (see Section 2.3 of the main paper for details) with 32 PP repo contexts each of size 768 tokens ($N = 32, l = 768$). Similar to Izacard & Grave (2021), we format each repo context with special tokens to mark the beginning of the surrounding context and the repo context, as well as for the name of the repo context (which is the same as the name of the PP taken from `https:`

`//github.com/shrivastavadisha/repo_level_prompt_generation`). We used `hole_context:` as prefix for surrounding context, `rule_context:` as a prefix for PP repo context and `rule_name:` as prefix for PP repo context name. We used Adam (Kingma & Ba, 2015) optimizer with a learning rate 1e-5 and a warmup linear scheduler with 5000 warmup steps. We used gradient clipping with norm 1.0 and batch size of 1. Training was carried out on 2 NVIDIA A100 GPUs with memory of 80GB each. Each evaluation run was carried out on a single 32GB V100 GPU.

The BM25 and Random NN versions of RepoFusion were obtained by using the same training hyperparameters as above and initialized from the same finetuned CodeT5 checkpoint except that we found that a learning rate of 2.5e-5 and the setting $N = 63, l = 512$ works the best. As before, we used NT-Prior-Last strategy and a prefix only for the surrounding context and no prefixes for repo contexts. The RepoFusion model that was initialized from a pretrained CodeT5-base version was obtained by using the same set of training hyperparameters as our best RepoFusion model but a learning rate of 1e-4 worked the best.

For the purpose of review, we have put all the trained checkpoints at `https://huggingface.co/anonymousTheStackRepo/trained_checkpoints`.

### B.3 RETRIEVAL MECHANISMS

The BM25 repo contexts were obtained using the Okapi BM25 implementation with default parameters given by the pip package `rank-bm25 0.2.2 https://pypi.org/project/rank-bm25/`. The BM25 scores are calculated with the surrounding context being the query and full context from other files in the repository being the search documents. Random NN repo contexts used the procedure followed by Shrivastava et al. (2022) using CodeBERT (Feng et al., 2020) to obtain the representations (See Appendix C.3 for details).

### B.4 OTHER BASELINES

For CodeGen models, we used the models available on Hugging Face hub, i.e., Codegen-2B-multi `https://huggingface.co/Salesforce/codegen-2B-multi`, CodeGen-6B-multi `https://huggingface.co/Salesforce/codegen-6B-multi` and CodeGen-16B-multi `https://huggingface.co/Salesforce/codegen-16B-multi`. For SantaCoder and StarCoder we used the model available from `https://huggingface.co/bigcode/santacoder` and `https://huggingface.co/bigcode/starcoder`, respectively. We used special FIM tokens, i.e., `<fim-prefix>` for pre context, `<fim-suffix>` for post context and `<fim-middle>` to prompt for completing the target hole. Each of these models used the recommended tokenizers and completion length of 128 tokens.

## C ADDITIONAL RESULTS

### C.1 EFFECT OF REPETITION

In order to further assess the significance of diverse repo contexts, we conducted an analysis by repeating a PPC multiple times and using each repetition as a separate repo context. One can see from the right side of Table 1 that repeating the context from a single prompt proposal (prior, post, randomly chosen PP) has a negative impact on performance compared to using different repo contexts from multiple prompt proposals.

Table 1: Completion success rate with repeating different types of PPCs multiple times.

|         | Success Rate(%) |
|---------|-----------------|
| **Rand**    | 37.18±0.48 |
| **Prior**   | 50.69±0.50 |
| **Post**    | 54.64±0.50 |
| **NT-Rank** | 71.92±0.45 |

## C.2   APPENDING SURROUNDING CONTEXT

Table 2 shows the performance of RepoFusion when we do not append the surrounding context to each repo context. We see that the performance drops significantly for all strategies when compared to when the surrounding context is appended. It should be noted that for these experiments, we used our best RepoFusion model that is trained to take the concatenation of surrounding context and repo context as input. It is highly likely that a RepoFusion model trained to not append the surrounding context would suffer from much less performance drop.

Table 2: Completion success rate with and without appending surrounding context.

|  | without Surrounding Context | with Surrounding Context |
|---|---|---|
| **T-Rand** | 13.89±0.35 | 66.53±0.47 |
| **T-Rank** | 25.06±0.43 | 72.78±0.45 |
| **NT-Rank** | 15.57±0.36 | 73.60±0.44 |
| **NT-Prior-Last** | 17.18±0.38 | 74.82±0.43 |

## C.3   PERFORMANCE OF PRETRAINED CODET5

Table 3 shows the performance on the test set when we directly use the pretrained CodeT5-base and CodeT5-large models. For these experiments, we use the special token `<extra_id_0>` to prompt the completion of the target hole. We see that the performance of these pretrained models is quite low, thereby creating the need to finetune these models on Java repositories on the next-token prediction objective. We see from the top section of Table 2 in the main paper that the finetuning helps a lot.

Table 3: Completion success rate on the test set for pretrained CodeT5.

| Model | Size (#params) | Effective context length | Context type | Success Rate (%) |
|---|---|---|---|---|
| CodeT5-base | 0.22B | 512 | prior | 2.42 (0.04) |
| CodeT5-base | 0.22B | 2048 | prior | 3.94 (0.05) |
| CodeT5-large | 0.77B | 512 | prior | 4.56 (0.05) |
| CodeT5-large | 0.77B | 2048 | prior | 9.51 (0.07) |