# OpenReview forum: "RepoFusion: Training Code Models to Understand Your Repository"
_ICLR.cc/2024/Conference — Submitted to ICLR 2024_

### Official Review · Reviewer_Ar6w · 2023-11-01

**Soundness:** 3 good
**Presentation:** 3 good
**Contribution:** 3 good
**Rating:** 6
**Confidence:** 4

**Summary:**

This paper presents a technique to train models with software repository context with the goal to improve model's ability to complete single line code. The paper's key insight is to perform concatenation of context strings retrieved using prompt proposal function in the embedding space, so that the model can leverage larger effective context when performing the code completion tasks.

The paper explores four ways to compose contexts (T-rank, T-rand, NT-rank, and NT-prior-Last), three prompt proposal methods (Random-NN, BM25 and PPC), and train the 220M CodeT5 model. The results are quite insightful, showing that RepoFusion can indeed take advantage of context training to better solve code completion tasks.

**Strengths:**

The paper is quite simple and the idea of concatenating contexts as part of the encoder input isn't alien to us (as previously studied in QA domain), but the paper executes pretty well in the software repo setting:
1. Studies of different methods to combine repo contexts, and recommendations of PPC + ranked lists as the best performing methods.
2. Comparison with many prior baselines to show the effectiveness.
3. Dataset creation that can benefit the community.

**Weaknesses:**

Given the paper's main contribution is the application of relatively known methods to a different domain, I would expect a bit deeper studies to show domain insights. Concretely, I have the following concerns:
1. The construction of the dataset performs random hole sampling: does the author check with how likely a hole have highly similar snippets in other files of the repo? If this is the case, then the high performance improvement may comes from the ability to "copy" from similar context, and it could be quite obvious that baselines without access to contextual information would be handicapped.

2. The evaluation baseline lacks a variation where contexts are concatenated as part of the prompt. For example, in models like SantaCoder, CodeGen or StarCoder, we could consider adding top-k contexts (concatenation + truncation) as part of the prompt to ask for completion. This baseline would be more fair in the sense that they also have access to the similar information that RepoFusion model has access to. Without this comparison, it is hard to highlight whether the biggest gain comes from training or comes from access to contextual information. I believe such experiments would be low-cost enough (no extra training needed) to complete in a short amount time.

3. The paper lacks qualitative study analyzing types of mistakes made by different models. This would be related to my concern #1. I hope the authors can provide some insights on which cases do RepoFusion solve but other large models fail. It can also help explain effectiveness of different ranking methods: how does higher ranking contexts relate to the success of completion.

**Questions:**

I would like the authors to elaborate my concerns.

---

### Official Review · Reviewer_U4U4 · 2023-11-01

**Soundness:** 1 poor
**Presentation:** 4 excellent
**Contribution:** 2 fair
**Rating:** 3
**Confidence:** 3

**Summary:**

Transformer models for code can be trained to predict a missing piece of code given its surrounding context. Often, the information needed to predict this missing piece of code comes not only from the code in the same file, but also from other files in the code repository. A key question is how to choose and use relevant pieces of code from the entire repository to guide the language model in making a good prediction for the missing piece.

This paper proposes a solution to this problem by building on two prior approaches.

* The first is RLPG (Srivastava et al, 2022) which introduces 63 different types of “prompt proposals” based on information in the repository. These include things like method names, bodies, identifiers, string literals, etc.

* The second is Fusion-in-Decoder (Izacard & Grave, 2021) which combines external sources of information to aid a question-answering model. They use a transformer encoder model on each of the sources, and a decoder attends over the concatenated encodings.

This paper uses the Fusion-in-Decoder approach, with the “prompt proposals” from Srivastava et al (2022) forming the external sources. But these cannot just be used as-is, because there are too many sources of information. So the authors propose multiple schemes to select a subset of the information in the 63 types of prompt proposals.

The authors contribute a dataset curated from The Stack, consisting of code with “holes” and prompt proposals for each hole. They evaluate their algorithm on this dataset, and find that it performs better than some chosen baselines.

**Strengths:**

1. The problem is an interesting one, and quite relevant, because IDE-based language models like Copilot are increasing in popularity, and they do need to learn to use repository context effectively.

1. The naive approach to solve this problem is to ask a language model to predict the missing tokens given surrounding context _within the same file_. The authors show that their algorithm performs much better than this naive approach, even using models that are much larger. This is a good empirical result.

**Weaknesses:**

1. I’m a little puzzled at the lack of comparison with _RLPG itself_, given that this paper references that approach so frequently, and that that approach has a similar motivation (although the downstream task might be slightly different). Isn’t it possible to compare against that approach directly, for instance, by using the RLPG classifier to select the best prompt proposal, and then appending this along with the rest of the context? If I understand correctly, your approach uses a fixed ranking among the different prompt proposals, as opposed to dynamically choosing the best proposal based on the target hole.

1.  Similarly puzzled at lack of comparison with RepoCoder despite it being cited multiple times. I recognize that I might be missing something here (maybe these are subsumed by your approach, or maybe they are not applicable here for some reason?), but this definitely needs clarification.

1. This approach is a simple combination of two existing approaches with some chopping and truncation to cut the prompt proposals to size. But the specific method by ranking prompt proposals according to a pre-fixed ordering and throwing away tokens outside a fixed-size window feels clumsy and not principled.

**Questions:**

1. See the Weaknesses section - is it possible to compare directly against other approaches? Or is there a specific reason why this comparison would be impractical?

1. Since StarCoderBase is actually trained on The Stack, is it true that it’s seen all the files in your evaluation dataset before? If so that could contribute to its superior performance, and would be worth a mention.

---

### Official Review · Reviewer_hziX · 2023-11-02

**Soundness:** 3 good
**Presentation:** 3 good
**Contribution:** 1 poor
**Rating:** 3
**Confidence:** 4

**Summary:**

Code completion models often face challenges when completing code that involves proprietary APIs not included in their training data. To overcome these limitations, the research introduces RepoFusion, a training framework designed to learn how to blend various relevant contexts from code repositories. This approach enables more precise and context-aware code suggestions, with a primary focus on single-line code completion—a task that mirrors real-world situations where users edit existing files in an integrated development environment (IDE). RepoFusion utilizes the Fusion-in-Decoder architecture to merge these contexts. Each repository context is combined with its surrounding context and encoded independently. Subsequently, the decoder simultaneously considers these concatenated encoded representations to generate predictions for filling in code gaps. The study demonstrates that RepoFusion, despite its smaller size, can outperform several larger models and even come close to matching the performance of models trained with more complex objectives. Furthermore, the paper conducts comprehensive ablation studies, shedding light on the crucial factors that influence RepoFusion, including the nature, length, and number of repository contexts, as well as other training configurations. To support future research in this area, the authors release Stack-Repo, a dataset featuring 200 Java repositories with permissive licenses and nearly deduplicated files, enriched with three types of repository contexts.

**Strengths:**

- The topic is timely and well motivated.
- This paper proposed a repository-level training method that works well on hole completion tasks.
- The writing is clear and self-contained. The paper presents experimental results to support the claims.
- Authors open source dataset to facilitate future research.

**Weaknesses:**

- The novelty of the work is thin. At a high level, the work combines repository-level prompt generation and Fusion-in-Decoder (FiD) approaches. As a result, it is hard to justify this work as an ICLR paper.
- The paper predominantly assesses the model's effectiveness in whole completion (single-line completions), which restricts its capacity to showcase its usefulness in real-world coding scenarios. To bolster its robustness and adaptability, it would be beneficial to subject RepoFusion to a broader spectrum of coding tasks, encompassing multi-line code completion [1].
- In the evaluation, no baseline model sees repo contexts. As a result, it is hard to see the value of the RepoFusion approach.

[1] Repocoder: Repository-level code completion through iterative retrieval and generation.

**Questions:**

- Is RepoFusion applicable to any programming language?
- Can the authors do some more analysis to evaluate why and how different types of hole completions benefit from their approach?
- T-Rank truncates the prompt proposals at L tokens. Would it be better to perform a syntax-aware truncation, so that e.g. entire span of tokens corresponding to a method signature, or an entire docstring is included in the repo context?
- In Table 2, does post+prior indicate code infilling tasks? It would be great if these details were explained in the table caption.
- Is it possible to set up baselines where LMs see relevant repo context? For example, [1] could be used as a baseline for the evaluation dataset used in this work. Currently, (as per my understanding) the baselines in Table 2 do not see any repo contexts.

[1] Repocoder: Repository-level code completion through iterative retrieval and generation.

---

### Meta-Review · Area_Chair_YVCU · 2023-12-14

**Metareview:**

This paper explores combining fusion-in-decoder (Izacard & Grave 2021) with repo-level context (RepoCoder, Zhang et al. 2023) and prompting (RLPG, Srivastava et al. 2022) for better code completion when leveraging repository contents. The novelty and new insights of the paper is limited. The base model was trained on TheStack, which the authors used for getting completion evaluation. The authors were not active in the discussion period. Overall, the paper falls short of the high bar for publication at ICLR.

**Justification For Why Not Higher Score:**

N/A

**Justification For Why Not Lower Score:**

N/A

---

### Decision · Program_Chairs · 2024-01-16

Reject